

# Differing responses of red abalone (*Haliotis rufescens*) and white abalone (*H. sorenseni*) to infection with phage-associated *Candidatus* Xenohaliotis californiensis

Ashley Vater[1], Barbara A. Byrne[2], Blythe C. Marshman[3], Lauren W. Ashlock[3] and James D. Moore[3,4]

[1] Integrative Pathobiology Graduate Group, University of California, Davis, Davis, United States of America

[2] Pathology, Microbiology, Immunology, School of Veterinary Medicine, University of California, Davis, Davis, United States of America

[3] Shellfish Health Laboratory, California Department of Fish and Wildlife, Bodega Bay, United States of America

[4] Karen C. Drayer Wildlife Health Center, School of Veterinary Medicine, University of California, Davis, Davis, United States of America

Corresponding author
Ashley Vater, awvater@ucdavis.edu

## ABSTRACT

The Rickettsiales-like prokaryote and causative agent of Withering Syndrome (WS)—*Candidatus* Xenohaliotis californiensis (*Ca.* Xc)—decimated black abalone populations along the Pacific coast of North America. White abalone—*Haliotis sorenseni*—are also susceptible to WS and have become nearly extinct in the wild due to overfishing in the 1970s. *Candidatus* Xenohaliotis californiensis proliferates within epithelial cells of the abalone gastrointestinal tract and causes clinical signs of starvation. In 2012, evidence of a putative bacteriophage associated with *Ca.* Xc in red abalone—*Haliotis rufescens*—was described. Recently, histologic examination of animals with *Ca.* Xc infection in California abalone populations universally appear to have the phage-containing inclusions. In this study, we investigated the current virulence of *Ca.* Xc in red abalone and white abalone at different environmental temperatures. Using a comparative experimental design, we observed differences over time between the two abalone species in mortality, body condition, and bacterial load by quantitative real time PCR (qPCR). By day 251, all white abalone exposed to the current variant of *Ca.* Xc held in the warm water (18.5 °C) treatment died, while red abalone exposed to the same conditions had a mortality rate of only 10%, despite a relatively heavy bacterial burden as determined by qPCR of posterior esophagus tissue and histological assessment at the termination of the experiment. These data support the current status of *Ca.* Xc as less virulent in red abalone, and may provide correlative evidence of a protective phage interaction. However, white abalone appear to remain highly susceptible to this disease. These findings have important implications for implementation of a white abalone recovery program, particularly with respect to the thermal regimes of locations where captively-reared individuals will be outplanted.

## INTRODUCTION

Abalone are iconic benthic invertebrates that contribute to ecological health of northern Pacific coast kelp forests and serve as a food source for endangered sea otters *Enhydra lutris*. California's wild abalone fishery flourished from the 1950s–1980s but was decommercialized in response to population declines from overexploitation followed by disease (*California Department of Fish & Wildlife, 2005*). Farmed abalone is of increasing economic significance; in 2008, it was estimated that over 129,000 metric tons of farmed abalone was supplied to the world market (*Cook, 2016*).

Withering syndrome (WS) was first reported in mid-1980's at the Channel Islands, California and decimated black abalone—*Haliotis cracherodii*—populations (*Haaker et al., 1992*). The infection causes reduced feeding behavior and nutrient absorption; animals wither as they lose body mass through catabolism of the foot muscle (*Gardner et al., 1995*). The causative agent of the disease, *Candidatus* Xenohaliotis californiensis (*Ca.* Xc) is a member of the Order Rickettsiales of the Alphaproteobacteria (*Friedman et al., 2000*). Analysis of five genes (16S rRNA, 23S rRNA, *ftsZ, vVirB11,* and *vVirD4*) suggested that *Ca.* Xc is most closely related to the *Neorickettsia* genus and is the most ancestral form of the *Anaplasmataceae* family studied to data (*Cicala et al., 2017b*). Transmission of *Ca.* Xc appears to be fecal-oral (*Moore et al., 2001*). *Ca.* Xc infects the luminal epithelium of the posterior portion of the esophagus (PE) and digestive gland (*Moore et al., 2001*). The *Ca.* Xc bacterium forms large oblong inclusions in the digestive tract epithelium, which are easily identifiable in hematoxylin- and eosin-stained tissue sections (*Friedman et al., 2000*).

In red abalone, *H. rufescens*, exposure to warm water events in the presence of *Ca.* Xc exacerbates morbidity and mortality in two synergistic ways: it reduces the nutritional content of their primary food source, and is associated with elevated pathogen burdens (*Vilchis et al., 2005*). Trends of increasing frequency and intensity of Pacific ocean warming El-Nino events correspond to dramatic reductions in giant kelp densities (*Tegner et al., 1996*), and nitrogen nutrient concentration in seawater is inversely related to temperature (*Tegner et al., 2001*). Furthermore, ocean warming trends coincide with *Ca.* Xc disease outbreaks (*Harvell et al., 1999*). The results of laboratory studies using juvenile farm-raised red abalone showed that *Ca.* Xc has relatively little effect on the health of abalone held at temperatures of approximately 14 °C, while animals held in water approximately 18 °C suffer high mortality rates in association with higher *Ca.* Xc body burdens (*Braid et al., 2005*; *Moore, Robbins & Friedman, 2000*; *Rosenblum et al., 2005*; *Vilchis et al., 2005*). *Braid et al. (2005)* demonstrated that clinical signs of withering syndrome are not solely due to warm water stress..

White abalone–*H. sorenseni*–health and fitness is best supported by a consistent 14 °C seawater environment as determined by optimization of captive breeding methods (*Leighton, 1972*; *Rogers-Bennett et al., 2016*). Increased seawater temperatures are associated

with adverse health effects in white abalone that translate to population instability. For example, white abalone spawning success rates are significantly decreased in warm water (20 °C). Ocean warming events have the potential to powerfully exacerbate disease, particularly in the sensitive white abalone species, while simultaneously reducing general animal fitness. White abalone were the first marine invertebrate species to be recognized as federally endangered. Outplanting animals bred and raised in a captive rearing program is considered the key restoration approach to increase densities quickly enough to reduce the probability of extinction (*Rogers-Bennett et al., 2016*; *Stierhoff et al., 2014*). White abalone are highly susceptible to WS (*Friedman et al., 2007*). This and globally increasing sea water temperatures may challenge future outplanting efforts.

After the catastrophic population declines reported in the early 90's, wild black abalone survival rates began improving in 1996 which could not be fully attributed to ocean cooling trends that repress expression of WS (*Chambers et al., 2005*; *Chambers et al., 2006*). Further, anecdotal observations from California red abalone farmers beginning in 2006 indicated that the incidence and severity of WS had diminished; this change correlated with the appearance of a novel bacteriophage hyperparasite associated with *Ca.* Xc (*Crosson et al., 2014*; *Friedman & Crosson, 2012*). This bacteriophage was first described in farmed red abalone from Cayucos, California examined in 2009 (*Friedman & Crosson, 2012*). Its presence was visualized by histology as morphologically distinct inclusions and transmission electron microscopy (TEM) confirmed the presence of phage particles in several studies. These distinct pleomorphic inclusions, independent from confirmatory TEM, are currently recognized as representing phage-containing *Ca.* Xc (*Brokordt et al., 2017*; *Crosson et al., 2014*; *González et al., 2014*). Further characterization suggests this phage is a member of the *Siphoviridae* family and employs a lytic life cycle, which results in lysis of the host cell and replicates its genetic material separately from that of the host (*Cruz-Flores & Cáceres-Martínez, 2016*).

Bacteriophages are more abundant than any other marine biological entity; however, there are only a relatively small number of published genomes. These likely fail to capture the high diversity of marine phages (*Perez Sepulveda et al., 2016*). In April 2018, the annotated genome of the *Ca.* Xc phage was published and represents the first phage of a marine rickettsial-like organism to be sequenced; the identified open reading frames had low levels of similarities to other known biological entities (*Cruz-Flores et al., 2018*).

While *Friedman et al. (2014a)* and *Friedman et al. (2014b)* demonstrated significant improvement in black abalone survival when challenged with phage-containing *Ca.* Xc, the mechanism by which the phage reduces pathogenicity is unclear. Complex host-pathogen-phage interactions and resulting selective pressures, which increase bacterial resistance to the phage and simultaneously diminished the bacteria's virulence, have been observed in other microbial systems (*León & Bastías, 2015*). Evidence supports prophage-associated changes in virulence in other Rickettsiales (*Masui et al., 2000*). Mechanisms for variation in *Ca.* Xc virulence unrelated to phage have been explored in recent years. A preliminary study investigating effects of breeding found that there is a genetic component to Ca. Xc infection susceptibility (*Brokordt et al., 2017*). Interbreeding between abalone species has been show to transfer susceptibility to WS (*González et al., 2014*). Heritable and environmental

variables may also influence the abalone host microbiome, which is generating interest as a possible factor associated with abalone health and resistance to withering syndrome (*Cicala et al., 2017a*; *Connelly, Horner-Devine & Friedman, 2012*).

In this study, we aimed to elucidate changes in *Ca.* Xc virulence in association with phage presence on WS expression in red and white abalone species under thermal conditions that are known to either enhance or retard expression of the disease.

## MATERIALS AND METHODS

### Abalone and life support

This experiment was conducted in the California Department of Fish and Wildlife's Pathogen Containment Facility at the UC Davis Bodega Marine Laboratory in Bodega Bay, California. Red abalone, approximately 2.2 cm in shell length, were purchased from an abalone farm in Goleta, California. White abalone, approximately 1.8 cm in shell length, were donated from the UC Davis Bodega Marine Laboratory's White Abalone Recovery Project. Prior to challenge, feces from all tanks were tested for 16S rRNA *Ca.* Xc genes by quantitative PCR (qPCR) following a validated protocol (*Friedman et al., 2014b*). Although fecal qPCR is treated only as a proxy for live pathogen, it has been shown to be the most sensitive assay for *Ca.* Xc detection (*Friedman et al., 2014b*). To support the absence of infection, five animals from both the red and white groups were sacrificed for histologic examination. Animals were supplied with a combination of wild-collected kelp (*Macrocystis pyrifera*) and cultured dulse (*Palmaria palmata*) two to three times per month throughout the experiment. Because *Ca.* Xc is known to be present in local abalone, feed was soaked in freshwater for at least 5 min prior to distributing to the tanks; our ongoing unpublished observations have demonstrated that this is sufficient to inactivate residual *Ca.* Xc that may be present on algal feed (CDFW unpublished observations). All tanks received constant 20-$\mu$m filtered, aerated, UV-irradiated, flow-through seawater.

### Experimental design

This study was constructed as a fully nested design with tanks nested within temperatures and *Ca.* Xc exposure challenge and abalone nested within tanks. One hundred ninety-two red and 192 white abalone, ~2 cm in length, were randomly and evenly distributed into either of the two treatment groups (exposed), or the control group (Fig. 1).

To avoid cross-contamination, we spatially organized the groups in lieu of random placing. Each treatment group was comprised of eight 3.8-L tanks, with eight animals housed in each tank. Two groups, one exposed to *Ca.* Xc and the other unexposed, received elevated temperature seawater (approximately 18.5 °C); a second *Ca.* Xc-exposed group received ambient water (approximately 13.6 °C). Temperature was measured hourly by automated temperature recorder placed in one tank per treatment group.

### Exposure to *Ca.* Xc

To initiate *Ca.* Xc exposure, inflowing seawater was routed through 11-L conical header tanks with farmed red abalone prior to supplying the experimental tanks. The two exposed groups of each species received effluent water from a header tank containing eight farmed

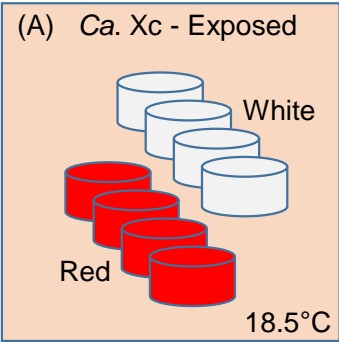
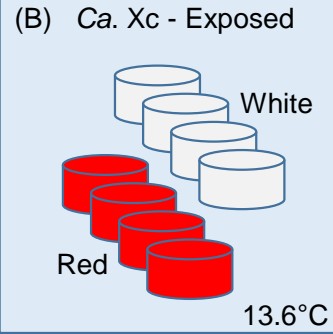
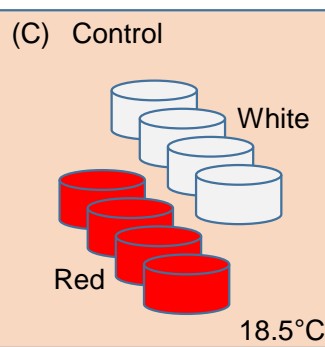

**Figure 1** Experimental set up: Experimental variables are illustrated as follows: (A) *Candidatus* Xenohaliotis californiensis (*Ca.* Xc) exposed, 18.5 °C; (B) *Ca.* Xc exposed, 13.6 °C; (C) Control, 18.5 °C. Experimental units–tanks–are represented as cylinders; red or white fill represents those stocked with red or white abalone respectively. Elevated (18.5 °C) and ambient (13.6 °C) seawater flowed from header tanks holding infected or uninfected animals for the first 161 days to transmit *Ca.* Xc.

red abalone, each approximately 123 gm in weight, from a population shown by histology to be infected with *Ca.* Xc and its phage. The control groups of red abalone and white abalone were headed by a tank holding 50 farmed red abalone, approximately 2.4 gm in weight; these animals' feces tested negative for *Ca.* Xc by PCR. The source water was directed through the headers through day 161 to ensure *Ca.* Xc exposure. *Moore et al. (2001)* showed that infection in red abalone was 100% after 111 days of *Ca.* Xc exposure to a header tank with infected red abalone at 18.5 °C (prior to appearance of the *Ca.* Xc phage).

### Sampling schedule and processing

At selected time points (days 0, 62, 126, 161, 265, 343) all animals in the experiment were weighed and measured for shell length. A body mass Condition Index (CI) was calculated as: $\frac{Total\ Weight\ (gm)}{Shell\ Length\ (cm)^3}$. At day 161, header tanks were removed and two randomly selected animals per tank were sacrificed and tested for *Ca.* Xc infection by qPCR from DNA extracted from post-esophagus (PE) tissue samples. Additionally, at day 161, six white abalone, three animals each from the heated and ambient exposed groups, were randomly selected and sacrificed for histological confirmation of transmission of *Ca.* Xc and its phage. At day 343, all surviving experimental animals were processed for analysis of infection in PE and digestive gland tissues by both qPCR and histology. For qPCR, PE tissue (∼30 mg) was excised from sacrificed animals and DNA extractions were performed using a DNeasy Blood and Tissue Kit (QIAGEN Germantown, MD) following the manufacturer's protocol for pathogen detection. Tissue samples were processed for histology as previously described (*Moore et al., 2001*). Davidson's-fixed (*Shaw & Battle, 1957*), hematoxylin- and eosin-stained 5 μm paraffin tissue sections containing PE and digestive gland were prepared from sacrificed animals. After termination of the experiment, slides were blindly assessed for presence/absence of *Ca.* Xc inclusions, and the inclusions present were categorized as having morphologies indicating phage infection (phage-containing) or lack of infection (classical).

### Fecal bacterial sampling regimen and processing

Feces were collected from each tank bi-monthly for qPCR analysis. Feces from four tanks per group were pooled for *Ca*. Xc 16S rRNA gene detection. Fecal samples were weighed and frozen at $-20\,°C$ upon collection until analysis. DNA from fecal samples ($\sim$250 mg) was extracted and purified with a QIAamp DNA Stool Mini Kit (QIAGEN) according to the manufacturer's 'Isolation of DNA from Stool for Pathogen Detection' protocol. DNA obtained was eluted in 200 µl volumes and stored at $-20\,°C$ until analysis.

### Quantitative PCR assays for *Ca*. Xc

We monitored *Ca*. Xc gene presence in post-esophagus and fecal samples using the methods developed and validated by *Friedman et al. (2014a)* and *Friedman et al. (2014b)*. Standard curves were constructed using PCR product of the WSN1 primers: WSN1 F (5′AGTTTACTGAAGGCAAGTAGCAGA3′) and WSN1R (5′TCTAAC TTGGACTCATTCAAAAGC3′) and the P16RK3 plasmid (*Friedman et al., 2014b*). Plasmid concentration was quantified by Qubit fluorometer (ThermoFisher Scientific, Waltham, Massachusetts). Assayed tissue and fecal samples were considered positive if the mean copy number per ng of genomic DNA in triplicate samples was equal to or greater than one, and reactions prior to normalization calculations had at least three gene copies—as convention of Minimum Information for Publication of Quantitative Real-Time PCR Experiments guidelines describes (Bustin et al., 2009). For reporting purposes, and to meet assumption of residual normality for statistical analysis, reaction copy numbers were normalized by input DNA (ng) and log transformed.

### Data analysis

A Cox proportional hazard model was used for survival analysis of treatment groups. A Chi-squared test was used to assess differences in survival of red abalone between this study and a historical study conducted prior to presence of the phage (*Moore, Robbins & Friedman, 2000*). Hazard ratio terminology refers to the likelihood of death associated with stratified variables: species, water temperature, *Ca*. Xc exposure. Variations in (1) condition index values and (2) qPCR data, were tested for significance by One-way ANOVA. Results of these models were assessed by *post hoc* Tukey comparisons; results are reported with estimates and standard deviations, describing the difference of the means and variance respectively from null hypothesis of the linear model. Residual errors from analysis were assessed for normality using the Wilk Shapiro test. Data from qPCR study was log-transformed. Both log-transformed qPCR data and condition index data was transformed by winsorization to meet the parametric test assumptions of normality. A test had a significant result if $p \leq 0.05$ (level of significance $\alpha = 0.05$). Statistical analysis was done with R version X R3.1.3 (*R Core Team, 2014*).

## RESULTS

### Transmission of *Ca*. Xc

All *Ca*. Xc-exposed groups showed fecal shedding of *Ca*. Xc DNA after header tank removal from the system (day 161), indicating effective transmission to the exposed red abalone
**Table 1** **Presence or absence of *Candidatus* Xenohaliotis californiensis (Ca. Xc) DNA in feces from experimental groups between day 0 and 342 by qPCR.** DNA extracted from tank feces was pooled by group and time period for PCR. Presence is defined as mean Ca. Xc gene copy number greater than or equal to three.

| Group: species, treatment, temperature | Days | | |
|---|---|---|---|
| | 0–113 | 126–236 | 251–342 |
| Red, *Ca*. Xc-exposed, ambient | − | + | + |
| White, *Ca*. Xc-exposed, ambient | + | + | + |
| Red, *Ca*. Xc-exposed, elevated | + | + | + |
| White, *Ca*. Xc-exposed, elevated | + | + | NA |
| Red, control, elevated | − | − | − |
| White, control, elevated | − | − | − |

**Notes.**

NA, no animals remain at this time period.

\+ greater or equal to 3 gene copies - less than 3 gene copies.

and white abalone under both temperature regimes. The unexposed tanks tested negative by feces PCR (Table 1).

Pathogen transmission to all *Ca*. Xc-exposed tanks was confirmed by tissue qPCR from animals sacrificed at the end of the experiment with the exception of the *Ca*. Xc-exposed, elevated temperature white abalone which all died prior to day 343. Abalone that died of natural causes during the experiment were frozen and PE was excised and screened by qPCR for *Ca*. Xc. Of these animals, 16% from the control groups produced very low levels of *Ca*. Xc gene amplification (<three copies/ng input DNA). The source of the potential contamination is unknown. Other measurements from fecal qPCR, tissue qPCR from sacrificed animals, and histology assessments did not indicate transmission of *Ca*. Xc. to the control groups.

## Analysis of survival

We examined survival in response to water temperature, abalone species, and *Ca*. Xc exposure (Fig. 2).

Our analysis indicated that white abalone exposed to *Ca*. Xc and elevated temperature were 10.9 times more likely to die than red abalone held in the same conditions (hazard ratio = 10.9; 95% CI [5.97–19.87]; $P < 0.001$). By day 251, all *Ca*. Xc-exposed white abalone in the elevated temperature treatment died, while red abalone held under the same conditions maintained a survival rate of 91%. Elevated temperature increased mortality risk for both red and white abalone 3.3 times that of the ambient treatment groups (hazard ratio = 3.3; 95% CI [1.1–10.2]; $P = 0.039$). Specifically, under elevated temperature, *Ca*. Xc-exposure increased the mortality risk of both species of abalone 12.5 times (hazard ratio = 12.5; 95% CI [1.6–96.0]; $P = 0.015$). Generally, under all conditions, white abalone had a risk of mortality that was 41 times that of red abalone (hazard ratio = 41.0; 95% CI [5.6–303.1]; $P < 0.001$).

The survival rate of red abalone in the current study, exposed to *Ca*. Xc with phage, was 36% higher than the previous experiment, in which animals were exposed to phage-free *Ca*. Xc (Table 2) (*Moore, Robbins & Friedman, 2000*). We evaluated survival at day 220,

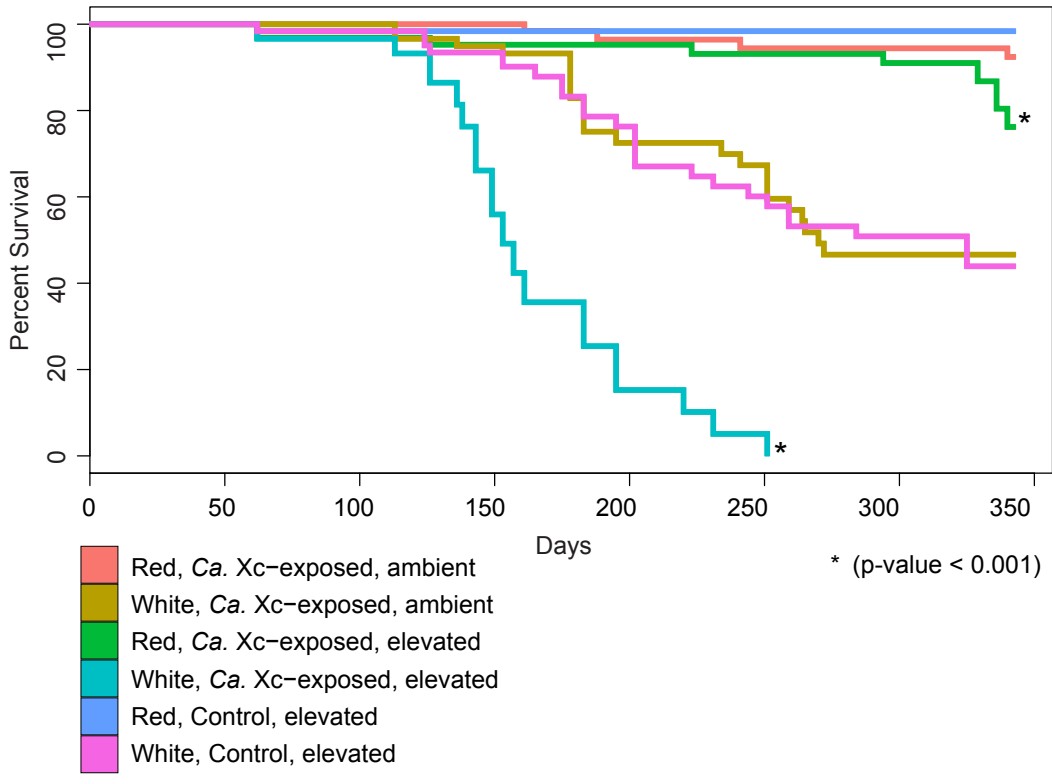

**Figure 2** **Percent survival of red and white abalone held in ambient water (13.6 °C) or at elevated temperature (18.5 °C) with and without *Candidatus* Xenohaliotis californiensis (*Ca.* Xc) exposure for 343 days; N = 64 for each group at day.** Each curve represents one of the six treatment groups, with variables: *Ca.* Xc exposure, seawater temperature, and species. We observe a significant difference between survival curves of red and white abalone held under elevated seawater temperature and *Ca.* Xc exposure conditions.

**Table 2** **Comparison of cumulative mortality rates between a 2000 study and the current study of thermal induction of withering syndrome in red abalone.**

| | Average seawater temperature | N | Cumulative mortality events at day = 220 | Percent survival |
|---|---|---|---|---|
| Current study | 18.5 | 46 | 4 | 91% |
| *Moore, Robbins & Friedman (2000)* | 18.5 | 30 | 10 | 67% |

immediately prior to termination day in the 2000 study (*Moore, Robbins & Friedman, 2000*). Between Days 1 and 220, in both experiments, the average water temperature was 18.5 °C (*Moore, Robbins & Friedman, 2000*). Both experiments used farm-origin juvenile red abalone. The 60 animals in *Moore, Robbins & Friedman*'s (*2000*) study averaged 8 cm in length and were selected from a farmed population with a known low-intensity *Ca.* Xc infection; however, this population did not express clinical symptoms of WS (*Moore, Robbins & Friedman, 2000*). Comparison of survival in the historical (phage-free) and

current (phage-containing) experimental data by Chi-squared test shows a significant difference (X-squared $= 25.37, df = 1, P < 0.001$) favoring survival in the presence of phage.

## Body condition indices

Assessment of animal health—as a function of weight, normalized by length—over time showed that red abalone remained heathy under all experimental conditions, unlike their white abalone counterparts (Fig. 3).

While data were collected at additional time points, we focused on three—beginning, mid, and end—for visual clarity. At the outset of the experiment (Day 0), the only group that showed a significant difference in CI values was the white, ambient group destined for *Ca.* Xc exposure, with a greater condition index values than the other five groups (Estimate 0.008, Std. Error 0.002, $t$-value $= 3.950, P < 0.001$). Mid-way through the experiment (Day 161) all three groups of white abalone overall had lower mean condition index values than their red counterpart groups. Statistical analysis showed significant differences between control, elevated temperature red and white groups (Estimate $-0.009$, Std. Error 0.003, $t$-value $= -3.497, P < 0.007$) with somewhat greater differences between the exposed red and white groups in both elevated temperature (Estimate $-0.020$, Std. Error 0.003, $t$-value $= -6.003, P < 0.001$) and ambient (Estimate $-0.017$, Std. Error 0.003, $t$-value $= -6.459$, $P < 0.001$) seawater treatments. However, no differences were observed between elevated temperature and ambient white, *Ca.* Xc-exposed groups (Estimate $-0.008$, Std. Error 0.003, $t$-value $= -2.386, P = 0.162$) more than half of the exposed elevated temperature white abalone had died prior to this time point. Based on visualization of longitudinal changes, white abalone fared worse under all treatments; *Ca.* Xc exposure appeared to be associated with increased withering in white abalone, while red abalone did not appear to decrease in body condition in response to *Ca.* Xc exposure. At the end of the experiment (day 343), white and red abalone under the same treatment of *Ca.* Xc-exposure in ambient seawater showed significant differences in body condition (Estimate $-0.024$, Std. Error 0.004, $t$-value $= -5.823, P < 0.001$). However, we observed significant, yet smaller differences between the control groups (Estimate $-0.013$, Std. Error 0.004, $t$-value $= -3.242, P < 0.0121$). There was not a significant difference in white abalone body condition between the control and *Ca.* Xc-exposed, ambient, groups (Estimate 0.004, Std. Error 0.005, $t$-value $= 0.955$, $P = 0.872$); additionally, there was no difference between red abalone in the control and the exposed, elevated temperature groups (Estimate 0.001, Std. Error 0.003, $t$-value $= 0.369, P = .996$).

## *Candidatus* Xc prevalence and infection intensity

*Candidatus* Xc 16S rRNA gene copy numbers obtained by qPCR from PE tissue can serve as a proxy for bacterial burden, and were assessed at days 161 and 343 (Fig. 4).

The qPCR data from tissue samples taken on day 161 were too skewed to transform such that residuals met normality assumptions for appropriate parametric statistical analysis. However, visualization of data trends suggests that at day 161, elevated temperature resulted in higher pathogen tissue burdens in white abalone (Fig. 4); the mean *Ca.* Xc gene copy number per ng DNA from tissue samples of exposed white abalone in the elevated

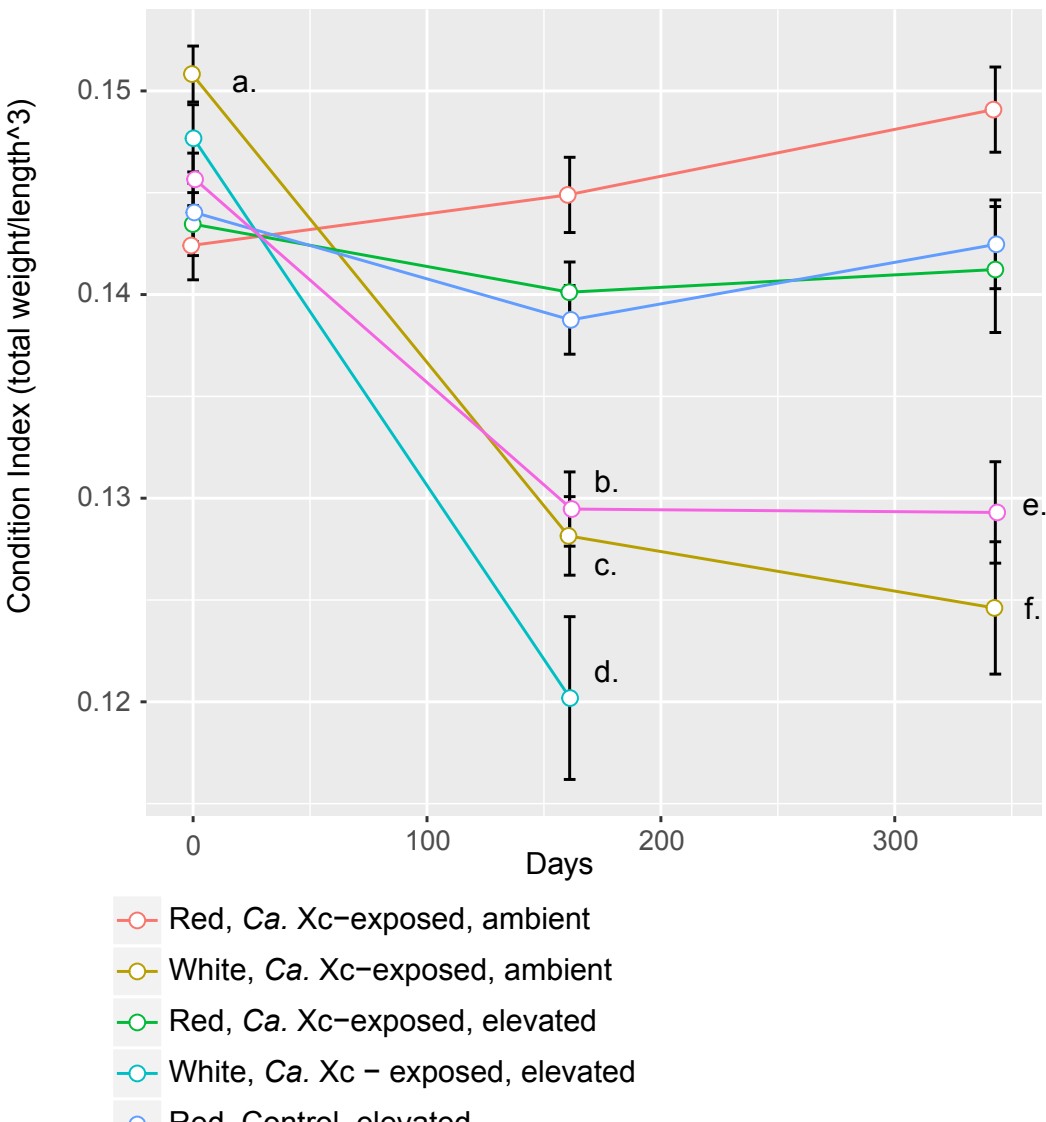

**Figure 3** **Longitudinal plot of mean values in Condition Index (CI), calculated by total abalone weight divided by shell length cubed, over time of experimental groups.** Error bars represent standard error of the mean. (A) Initial CI values of white abalone, destined for *Ca.* Xc exposure were significantly greater than the other groups. (*p*-values < 0.05) (B) CI values of white, control abalone were significantly lower than red, control abalone (*p*-value = 0.007) (C) CI values of white, *Ca.* Xc-exposed, ambient abalone were significantly lower than red, *Ca.* Xc-exposed, ambient abalone (*p*-value < 0.001) (D) CI values of white, *Ca.* Xc-exposed, elevated abalone were significantly lower than red, *Ca.* Xc-exposed, elevated abalone (*p*-value < 0.001) (E) CI values of white, control abalone were significantly lower than red, control abalone (*p*-value = 0.007) (F) CI values of white, *Ca.* Xc-exposed, ambient abalone were significantly lower than red, *Ca.* Xc-exposed, ambient abalone (*p*-value < 0.001).

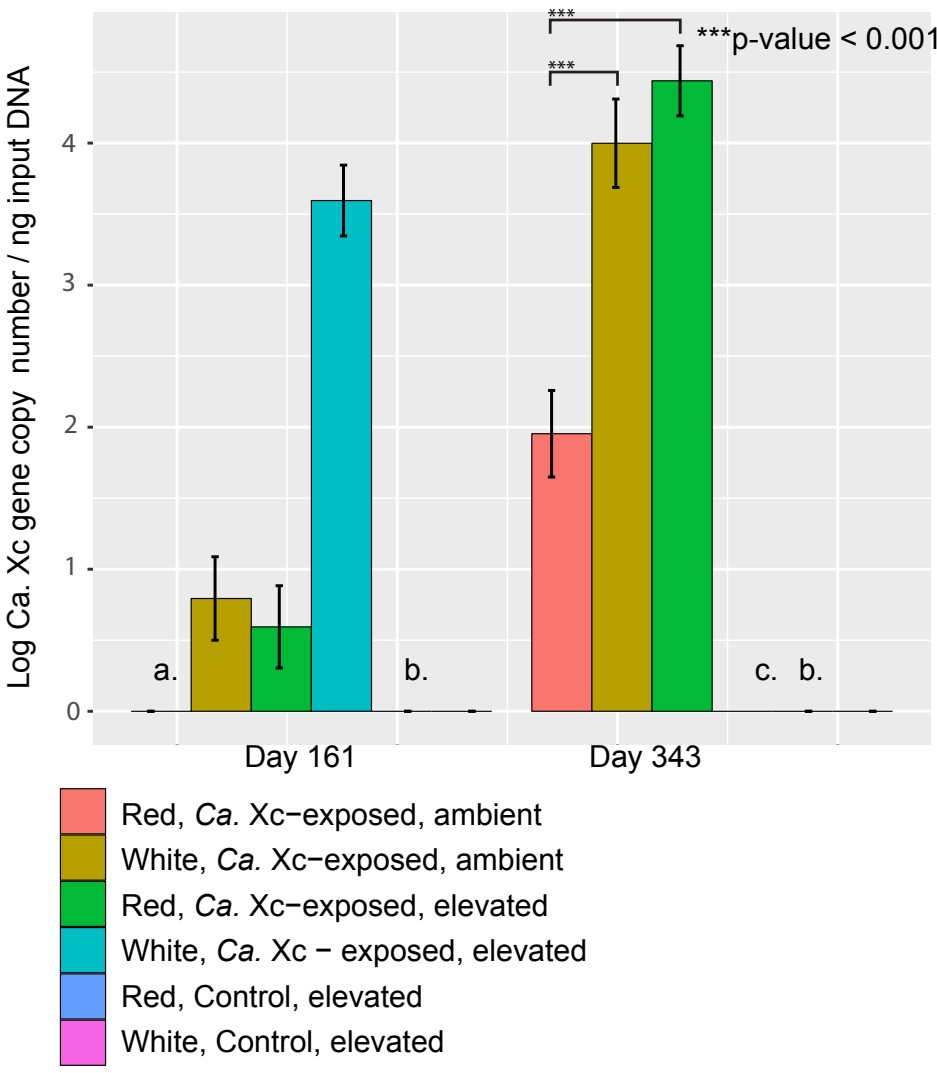

**Figure 4** **Log transformed qPCR-derived** *Candidatus* **Xenohaliotis californiensis gene copy numbers from PE tissue at days 161 and 343.** (A) At day 161, no *Ca.* Xc genes amplified in the red, ambient group (B) No *Ca.* Xc genes amplified in the control groups (C) Prior to day 343, all abalone in the white, *Ca.* Xc-exposed, elevated group died.

temperature regimen was 4,248; 326 times greater that of their red counterparts. Notably, white animals sacrificed at this time point were survivors—60% of their cohort already died. There was no *Ca.* Xc amplification in the red, exposed ambient group at day 161. At day 343, red abalone in the exposed, ambient group had significantly lower pathogen gene numbers than (a) the exposed elevated temperature red group (Estimate 2.485, Std. Error 0.3167, $t$-value =7.845, $P < 0.001$) and (b) their exposed, ambient white counterparts (Estimate 2.045, Std. Error 0.3167, $t$-value =6.475, $P < 0.001$). No *Ca.* Xc DNA was amplified from PE tissue samples of sacrificed animals in the unexposed groups at either time point.

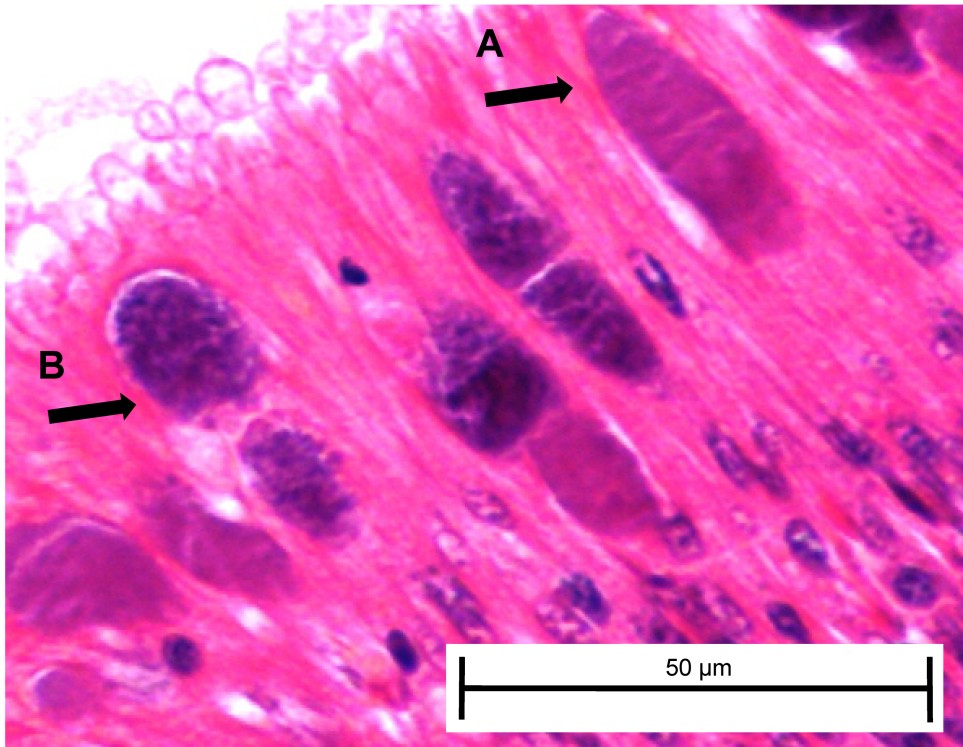

**Figure 5** *Candidatus* **Xenohaliotis californiensis inclusions within posterior esophagus epithelia from white abalone held at 18.5 °C and** *Ca.* **Cx exposed at 161 days.** Arrows indicate (A) classical inclusions, (B) Phage-containing variant inclusions. Haematoxylin and eosin. Bars, 50 μm.

At day 161, three white abalone from each of the exposed ambient and exposed elevated temperature groups were sacrificed and used for histopathology to corroborate transmission data from fecal and tissue qPCR samples and determine whether any *Ca.* Xc inclusions present included those with morphology indicating phage infection. For the first time, we morphologically identified the phage-containing inclusions in white abalone (Fig. 5).

Histological examination of day 343 samples showed the presence of *Ca.* Xc classical inclusions and phage-containing inclusions in the red, elevated temperature and white, ambient temperature groups (Table 3). In concordance with the qPCR data, we observed nearly three times as many classical inclusions and more than five times as many phage-containing inclusions in post-esophagus samples from the exposed red, elevated temperature group compared to the exposed white, ambient group. *Candidatus* Xc inclusions (both classical and phage-containing) in digestive gland tissue were only observed in the *Ca.* Xc-exposed, elevated temperature, red abalone group.

## DISCUSSION

The initial goal of this study was to investigate potential diminished pathogenicity of *Ca.* Xc in association with phage presence. However, the clear differences in expression of WS between white and red species is a key and unexpected finding. The white abalone control

**Table 3** Histological analysis of *Candidatus* Xenohaliotis californiensis - inclusion type prevalence in post esophagus and digestive gland tissue samples by group at day 343.

| Species, treatment, temperature condition | Post-esophagus | | | Digestive gland | | |
|---|---|---|---|---|---|---|
| | | Inclusion type[*] | | | Inclusion type[*] | |
| | N | Classical | Phage-variant | N | Classical | Phage-variant |
| Red, *Ca*. Xc-exposed, ambient | 34 | 0% | 0% | 54 | 0% | 0% |
| White, *Ca*. Xc-exposed, ambient | 14 | 29% | 14% | 19 | 0% | 0% |
| Red, *Ca*. Xc-exposed, elevated | 35 | 89% | 77% | 46 | 20% | 15% |
| White, *Ca*. Xc-exposed, elevated | 0 | NA | NA | 0 | NA | NA |
| Red, control, elevated | 27 | 0% | 0% | 62 | 0% | 0% |
| Red, control, elevated | 8 | 0% | 0% | 22 | 0% | 0% |

Notes.
[*]% of tissue samples with inclusion type.

group was adversely affected by the heated seawater—as observed by body condition and survival of the control group—perhaps contributing evidence for the frailty of the species. White abalone, exposed to *Ca*. Xc and held in 18.5 °C seawater, exhibited the highest mortality rate and most pronounced clinical signs of withering syndrome among the experimental groups, thus appearing to be highly susceptible to WS infection and disease, even with current phage-presence. However, under conditions that previously had been shown to exacerbate disease, the effects of WS from the current variant of *Ca*. Xc in association with the phage are reduced in red abalone relative to a similar study conducted more than a decade beforehand when no phage was evident.

Transmission of *Ca*. Xc was slowest in the red, ambient group based on fecal and tissue qPCR values. Variables that influenced fecal production and degradation, primarily feeding schedule, impaired normalization and thus interpretation accuracy of qPCR quantitative data, and therefore Table 1 summarizes only the presence/absence of *Ca*. Xc genes. Although, previous work has demonstrated PCR to be the most sensitive *Ca*. Xc detection method, it should be mentioned that due to assay sensitivity limitations, contamination in the control groups may have gone unidentified. In accordance with our results, slow infection progression in cold water has been show in previous studies (*Braid et al., 2005*; *Moore, Robbins & Friedman, 2000*). While the *Moore, Robbins & Friedman (2000)* study identified *Ca*. Xc inclusions in 90% of experimental animals held at ambient temperature (14.7 °C) at day 220, in the current study no inclusions were found by histology in the exposed red abalone held at 13.6 °C sacrificed on day 343. This finding from *Moore, Robbins & Friedman*'s (*2000*) study may be the result of 'natural' farm-associated transmission at a time point prior to the start of the study but it may also be indicative of a more virulent *Ca*. Xc strain. Additionally, the small histology subsample taken at day 161 from white abalone, held under ambient conditions did not have inclusions. We speculate that phage presence may also have extended the disease incubation period in the ambient temperature groups.

From weight and length data, we can infer that both the white and red abalone destined for *Ca*. Xc-exposure and elevated temperature treatment started with robust body condition, and thus were not at a disadvantage that would have predisposed them

to withering and death. The similarity between the elevated temperature, control and ambient exposed white groups' survival curves and condition index values supports the general sensitivity of these animals to adverse conditions and may also be a result of low genetic diversity (*Gruenthal & Burton, 2005*). The pronounced effects of temperature may be explained by white abalone's natural habitat; they are a deeper-dwelling species and experience less fluctuation in water temperature than other species that inhabit intertidal and shallower subtidal zones.

In order to examine the impact of the phage on *Ca.* Xc pathogenicity, it would of course have been ideal to directly compare the phage-containing and uninfected pathogens. However, currently this is not believed possible because classical inclusions are consistently accompanied by phage-containing ones. This has been recorded in recent histology assessments from abalone populations in Baja Mexico and Cayucos, San Nicolas Island, and Carmel California (*Cruz-Flores et al., 2016*; *Friedman et al., 2014a*) and has been observed in abalone from Bodega Head (J Moore, pers. obs., 2015). Consequently, we attempted to replicate a thermal induction study undertaken with red abalone prior to appearance of the phage (*Moore, Robbins & Friedman, 2000*). Comparing our results with those from that study strongly suggest that red abalone are better able to withstand withering syndrome now than in years previous to phage presence. Anecdotal observations from California red abalone farms support this conclusion (J Moore, 2002, unpublished data).

However, these differences may be the result of other changes in the system, particularly as red abalone have been challenged by the disease for nearly 30 years. For example, red abalone may have developed heritable traits that confer immunity, as suggested by *Brokordt et al. (2017)*. Alternatively, it is possible that *Ca.* Xc may have evolved to become less pathogenic through any number of mechanisms, including prophage; such a change cannot be elucidated by the experimental design of this study and thus we cannot decipher those mechanisms or causations. Recently, analysis of a subset of *Ca.* Xc coding genes showed an absence of genetic variation among samples taken from different geographic locations and from different abalone species; suggesting that genetic variability in the bacterium may have a limited contribution to the differing pathogenic effects we observe in this system (*Cicala et al., 2018*). A genome wide analysis might identify potentially attenuating virulence factors, which could be further investigated for population variation as explored in the (*Cicala et al., 2018*) study; however, whole genome sequence of *Ca.* Xc has yet to be published.

In this study red abalone exposed to the *Ca.* Xc showed no difference in body condition from their control counterparts, suggesting the limited pathogenic effects of the current variant *Ca.* Xc, or at a minimum the WS expression was significantly delayed. This is in contrast to the white abalone, for which we observed trends in body condition and survival that appear to be directly related to *Ca.* Xc gene copy numbers detected by qPCR. At the mid-point of the experiment, coinciding with the highest mortality rate and lowest body condition index values, *Ca.* Xc-exposed white abalone in the elevated temperature group also showed the greatest bacterial burden by qPCR. Despite any measured evidence in body shrinkage in the red groups, at the end of the experiment, *Ca.* Xc-exposed red abalone in the elevated temperature group had the highest pathogen gene copy and after day 300, the

*Ca.* Xc-exposed, elevated temperature red group's mortality rate appeared to increase; this might be indicative of disease expression after a nearly yearlong incubation period. The high level of detection of *Ca.* Xc DNA by tissue qPCR and histological changes at the end of the experiment could be associated with this apparent increase in mortality.

Our molecular data was supported by histological analysis. At day 343, *Ca.* Xc inclusions were only found in tissue samples from the two groups with the highest qPCR *Ca.* Xc gene amplification. Red abalone had a greater ratio of phage-containing inclusions to classical inclusions. However, it is difficult to assess and isolate the impact of temperature and species on our observations because we were only able to compare the red, elevated temperature group with the white, ambient group. While this study did not confirm phage particles by TEM, the highly-recognizable granular, pleomorphic inclusions have consistently revealed observable phage (*Cruz-Flores & Cáceres-Martínez, 2016*; *Cruz-Flores et al., 2016*; *Friedman et al., 2014a*). The mottled, complex morphology of the phage-containing inclusions has been interpreted as evidence of active phage-induced lysis (*Friedman & Crosson, 2012*). Inclusions with this morphology were observed in our experimental white abalone; this study is the first to document that white abalone are able to harbor the phage-containing variant of *Ca.* Xc.

## CONCLUSION

The results of this study have implications for restoration strategies to ultimately remove white abalone from the US Endangered Species list. The federal white abalone recovery plan concluded that outplanting of hatchery-produced animals must be the key restoration action for successful recovery of the species (*Team TWAR, 2008*). White abalone in warm water appear highly susceptible to the current variant of *Ca.* Xc, and outplanting efforts should take place in cooler water to minimize thermal enhancement of disease expression. The white abalone captive breeding program has recently introduced new wild-origin to its broodstock pool, which may increase genetic variation and render the progeny to be less sensitive to natural environmental stressors and possibly more resistant to the effects of *Ca.* Xc. Studying heritable variation in susceptibility of white abalone families would be highly informative to the captive breeding efforts. In the Southern California Bight, cooler water typically translates to deeper water but also certain geographic locations with strong upwelling such as San Miguel Island (*Erlandson et al., 2008*). Our findings with red abalone align with the anecdotal reports from California abalone farmers that disease caused by *Ca.* Xc has been much less frequent and severe since the phage has been observed. We conclude that the current form of *Ca.* Xc, with its phage present, is associated with improved health and survival in red abalone under conditions that have previously exacerbated the disease. However, the stability of this development is unknown. Further investigation of the genome and the *Ca.* Xc phage's phylogenetic relationships may shed light on the some of the host-pathogen-phage interactions and provide an explanation for the observed effects associated with the current pathogen variant. Future efforts may be directed towards whole genome sequencing of *Ca.* Xc and annotation of all mobile genetic elements of both the bacteria and the phage such that strains can be characterized

and better understood to elucidate the mechanisms associated with pathogenicity in this important marine system.

## ACKNOWLEDGEMENTS

We thank Chris Barker for statistical analysis of survival, Neil Willits for statistical consultation, and the White Abalone Recovery Project at the UC Davis Bodega Marine Laboratory for donating animals to this experiment.

### Funding

This work was supported in part by the Fisheries Branch, California Department of Fish and Wildlife; and The University of California Institute for Mexico and the United States (UC MEXUS) and the Consejo Nacional de Ciencia y Tecnologica de Mexico (CONACYT) (No. CN-14-14). There was no additional external funding received for this study. The funders had no role in study design, data collection and analysis, decision to publish, or preparation of the manuscript.

### Grant Disclosures

The following grant information was disclosed by the authors:
Fisheries Branch, California Department of Fish and Wildlife.
University of California Institute for Mexico and the United States (UC MEXUS).
Consejo Nacional de Ciencia y Tecnologica de Mexico (CONACYT): CN-14-14.

### Competing Interests

The authors declare there are no competing interests.

### Author Contributions

- Ashley Vater conceived and designed the experiments, performed the experiments, analyzed the data, prepared figures and/or tables, authored or reviewed drafts of the paper, approved the final draft.
- Barbara A. Byrne conceived and designed the experiments, analyzed the data, contributed reagents/materials/analysis tools, prepared figures and/or tables, authored or reviewed drafts of the paper, approved the final draft.
- Blythe C. Marshman and Lauren W. Ashlock performed the experiments, authored or reviewed drafts of the paper, approved the final draft.
- James D. Moore conceived and designed the experiments, performed the experiments, analyzed the data, contributed reagents/materials/analysis tools, prepared figures and/or tables, authored or reviewed drafts of the paper, approved the final draft.

### Data Availability

The R scripts and raw data are provided in the Supplemental Files.

## Supplemental Information

Supplemental information for this article can be found online at http://dx.doi.org/10.7717/peerj.5104#supplemental-information.

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
