# Peer review of "Differing responses of red abalone (Haliotis rufescens) and white abalone (H. sorenseni) to infection with phage-associated Candidatus Xenohaliotis californiensis"

_PeerJ, doi:10.7717/peerj.5104_

## Round 0.1 · original submission · Major Revisions

I agree with the reviewers’ comments, and request you reframe your manuscript in light of both sets of feedback. Phages may be solely responsible for the improvements you observe in your study, but there is currently insufficient proof. Please make it clear the phage is one potential reason why there have been recent improvements in abalone populations, and amend the title of your article to reflect this. As noted by reviewer 1, the R code associated with Figure 3 should be provided in full to allow others to replicate it from the data you provide. Please also note reviewer 2’s comments regarding reporting of statistical data.

I look forward to receiving your revised manuscript in due course, and thank you for submitting your article to PeerJ.

·

Basic reporting

1/. Basic Reporting:
• The manuscript is written in unambiguous and professional English. The manuscript does not require English language editing services.
• The manuscript’s introduction provides an overview to the Ca. Xc infection of abalones and the history and severity of the problem, including how changes in water temperature effect abalones. All of these are topics relevant to the study’s findings.
• The introductory section is sparse on the biology of phages. The review of Friedman et al. (2014), which is referenced within the introduction, does discuss the diversity of phages within oceans and phage lytic versus lysogenic lifecycles. These topics are relevant to the study’s findings.
• The structure and format of the manuscript is professional and correct.

Experimental design

2/. Experimental design
• The authors do note that this study builds upon other work, lines 294-296, where ‘they replicate a thermal induction study undertaken with red abalone prior to the appearance of the phage’. However, the work presented in this study is a self-contained unit of research that advances the scientific community’s understanding of Ca. Xc infection of abalones under a different conditions.
• While the study is well designed and conclusive in its findings (that there have been recent differences in Ca. Xc pathogenicity of abalones with changes in water temperature), the title of the manuscript is misleading. The study suggests the altered effect/reduction or elimination of Ca. Xc in abalones is caused by phages. At no point during the study are phages confirmed as present. The phenotypic alterations of the Ca. Xc (which are clearly observed) could be due to the past exposure of Ca. Xc strains to phages and not due to continued phage presence (for example, see Leon & Bastias (2015) “Virulence reduction in bacteriophage resistant bacteria”). Thus, the manuscript is more an examination of the current effects of Ca. Xc on red and white abalones (with phage as a potential confounding factor explaining the discrepancies between present and past observed differences).
• The work presented has been performed to a high technical standard, with statistical analysis performed to underpin the study’s observations.
• The raw data and R scripts for Figures 2, 4 and 5 allow for the reproduction of Figures within the manuscript. The R script for Figure 3 appears to be missing its start, and the relevant tables/objects cannot be loaded, thus the final image cannot be recreated without additional input. Therefore, the raw data for Figure 3 is also not verified.
• The study is well designed, even acknowledging some of its limitations (lines 291-292).

Validity of the findings

3/. Validity of the findings
• As stated previously, the study is well designed and conclusive that there are current differences in Ca. Xc virulence on abalones relative to previous years. The reduced mortality observed in abalones coincides with the appearance of a phage in Ca. Xc populations.
• However, no description of Ca. Xc populations before and after the appearance of this phage is presented nor is the phage characterised in silico or in vitro. Thus, any changes in Ca. Xc and phage populations are anecdotal and need to be treated more cautiously.
• The observed differences in abalone mortality may have been as a consequence of changes in bacterial populations, whereby an emergent lytic phage changed the Ca. Xc population. Alternatively, a new Ca. Xc strain emerged which carried a prophage. Scenarios such as these have not been sufficiently discussed as plausible and the authors claim an uncharacterised phage is responsible for their observations without sufficient evidence.
• While the study is of merit showing clear differences are observed in abalones with current Ca. Xc populations, there is too much emphasis on the phage being a sole driver of change and not sufficient explanation of future directions. Only the final sentence of the manuscript, lines 337-339, provides a positive outlook and future direction for controlling Ca. Xc decimation of abalone populations.
• The authors should consider discussing the need for a population examination of Ca. Xc, looking at strain level differences and potential attenuation of virulence between current populations and previous infective bacteria.
• The manuscript’s discussion also needs to build upon the need for characterising all mobile genetic elements, including phages, and also their encoded functions.
• Finally, the selective pressure exerted by continued Ca. Xc infection and mortality may have caused population shifts in abalones (perhaps more dramatically in red abalones than white) resulting in resistance. This is mentioned in the text, lines 327-330, where the authors note the introduction of genetic variation in abalones. Again, however, there is no discussion for the need to study abalone populations considering its endangered status and economic importance.

Additional comments

The study is of interest, showing recent changes in the virulence of Ca. Xc and thus increased survival of abalones. As global water temperatures increase, this will become an even more important topic. If presented correctly, this manuscript could be referenced frequently as a keystone upon which is built an example of how changes in oceanic conditions could lead to changes to phage, bacterial, invertebrate and vertebrate populations. However, in my opinion, too much weight is put on phages as the sole driver of the change in Ca. Xc. There needs to be a more cautious approach to explaining that phage may be the reason there has been an improvement in abalone populations. This is until Ca. Xc and phage populations are characterized in more detail. In addition, there are many avenues for future research that should be discussed to give a more complete picture to the severity of the problem faced.

·

Basic reporting

The manuscript is very well written and clear

Experimental design

The were some minor issues with the experimental design, nothing that cannot be accounted for in the interpretation of the results. See notes to the authors.

Validity of the findings

It is difficult to draw strong conclusions about the effects of the phage given your experimental design. This isn't critical, but I do think its best to reframe this manuscript. See notes to the authors.

Additional comments

I was very excited to review this manuscript. Abalone withering syndrome has led to dramatic population declines in California black abalone and contributed to the closure of the multi-species California abalone fishery. All of California’s abalone species appear to have differential resistance to the agent of withering syndrome, the Rickettsiales-like prokaryote Candidatus Xenohaliotis californiensis (Ca. Xc), and if infected with this pathogen, all abalone species also appear to demonstrate differential susceptibility to clinical signs of withering syndrome. Much of this fantastic body of work has been conducted by the authors here. Thus, this system has brought new insights into the ecology of marine diseases and the role of environmental conditions on disease pathology and spread, all within a multiple host community. Recent work has also identified a novel bacteriophage infecting Ca. Xc. There is a rich future ahead for innovative research on the ecology and pathology of marine diseases using this system as a model.

That said, I feel you tried to do too much with this dataset. The single biggest issue is that you cannot draw causal inference about the effects of the bacteriophage given the experimental design. Koch’s postulates apply to the effect of the bacteriophage on Ca. Xc. Given your experimental design, you cannot decipher whether presence of the bacteriophage led to decreased pathology, or increased immunity (and decreased pathology) led to the presence of the bacteriophage. I don’t think your inference is totally off, but I do think that both processes may be occurring. Understanding the relative contributions of 1) the interaction between abalone and Ca. Xc on phage dynamics and 2) the interaction between Ca. Xc and the phage on abalone health will be an exciting area for work in the future. Since we cannot yet culture either Ca. Xc or the phage, or reliably isolate the Ca.Xc from the phage and the phage from the Ca.Xc, this unfortunately has to remain a question for the future.

The second issue, and I don’t think this is huge deal, is the overall “frailty” of white abalone in your study, even in your unexposed control group (please replace the term “naïve” with “control” in your revision). I could be wrong, but I think two things are going on: first, white abalone are very susceptible to Ca. Xc, hence high mortality in both exposed groups. Second, you did not have a control group at ambient (cold) temperature, so I can’t make a strong inference here, but it seems white abalone are more sensitive to temperature independent of exposure to Ca.Xc. This makes sense given that they are presumptively a deeper water species. Given this though, your baseline hazard in your controls differed between the two species. Which is fine, but you should be clear about this in your interpretation of any results. A Cox model stratified by abalone species would apply well here.

I think it will be best for you to reframe this manuscript. Rather than focus on the phage, focus on the clear differences between the species – both in overall frailty and the effect of exposure to Ca.Xc. You can discuss the presence of the phage in response to your treatments, especially since you report the occurrence of the phage in white abalone (which is novel and exciting), but you should be clear about cause and effect. A clear presentation of differences between red and white abalone in WS pathology following exposure to Ca. Xc will be a valuable contribution to science. I am happy to help you reframe and revise this manuscript in any way.

Also, if you have PE or DG samples taken at the time of observed mortality, I think it will be very helpful to have qPCR and histology data from these time points.

Another minor issue is your source of exposure. A head tank full of abalone presumptively infected with Ca.Xc is well documented as a valid source of exposure. I imagine you placed abalone in the head tank of your control group to also control for exposure to everything else beyond Ca.Xc that is associated with abalone feces. The problem with doing this lies in using qPCR of feces to confirm exposure. The qPCR is a sensitive test but it is not definitive. You therefore cannot rule out some exposure to Ca.Xc in your unexposed control group. I don’t think this is a big deal since you didn’t see much Ca.Xc in your control animals over the course of your experiment, but be clear about all possible sources of exposure in your reporting.

And a minor reporting note. When reporting statistical tests such as ANOVA and the Cox model, please report full statistical results in the manuscript text rather than just P-values.

---

## Round 0.2 · accepted · Accept

As you will see from their comments below, both reviewers are happy with your revised manuscript and recommend its publication. I agree with their assessment of the revised document, and thank you for being so thorough in your revisions. I look forward to seeing your accepted manuscript published in the very near future.

# ·

Basic reporting

1/. Basic reporting:
• Possibly a few typos and commas which will be picked up by PeerJ’s proofing, but overall the manuscript is well written and very easy to follow. Thank you.
• Sufficient and informative references are provided.
• Images look well, and with the help of the figures legends, show clearly what is desired.
• Raw data and R scripts are provided and working correctly to reproduce all images. However, there are a few extra lines of code at the bottom of Figure_2’s script (which are not necessary to produce Figure_2). In order to run this extra few lines of code I needed to install and load extra packages (‘ggplot2’ and ‘survminer’). Just so you know.
• Overall, the results of the manuscript are ‘self-contained’ and not dependent on additional publications or knowledge to read and understand.

Experimental design

2/. Experimental design:
• While the experimental design has its limitations, these limitations are discussed. The overall study design is clearly described to independently reproduce the results observed.

Validity of the findings

3/. Validity of the findings:
• The study’s goal is clearly defined with the results and discussion addressing the current and future aims sufficiently. These results are important and novel and will be important for future implementation of best practices into abalone farming.

Additional comments

4/. General comments:
I am very pleased to have been involved in reviewing such a well prepared and clearly described experiment, looking at the effects of temperature and pathogens against abalones. I had concerns in the previous draft regarding the over-emphasis of the phage’s role in the observed effects. However, the authors have done a very good job in providing additional background information and also cautiously explaining other possibilities for their observed results. With these changes, I will support the publication of this manuscript without additional edits.
Thank you for taking the time to address my concerns, and best of luck with what appears to be many future avenues of research ahead.

·

Basic reporting

The manuscript is clear, well-written, unambiguous, sufficiently cited, professional, and self-contained with relevant results to the hypotheses tested.

Experimental design

The research question is well defined and rigorously investigated.

Validity of the findings

The data are robust and conclusions are well stated.